# Optical Solitons of the Generalized Nonlinear Schrödinger Equation with Kerr Nonlinearity and Dispersion of Unrestricted Order

Nikolay A. Kudryashov [1,2] 

1    Department of Applied Mathematics, National Research Nuclear University MEPhI (Moscow Engineering Physics Institute), 31 Kashirskoe Shosse, 115409 Moscow, Russia; nakudryashov@mephi.ru
2    National Research Center "Kurchatov Center", 1 Akademika Kurchatova Sq., 115409 Moscow, Russia

**Abstract:** The family of the generalized Schrödinger equations with Kerr nonlinearity of unrestricted order is considered. The solutions of equations are looked for using traveling wave reductions. The Painlevé test is applied for finding arbitrary constants in the expansion of the general solution into the Laurent series. It is shown that the equation does not pass the Painlevé test but has two arbitrary constants in local expansion. This fact allows us to look for solitary wave solutions for equations of unrestricted order. The main result of this paper is the theorem of existence of optical solitons for equations of unrestricted order that is proved by direct calculation. The optical solitons for partial differential equations of the twelfth order are given in detail.

**Keywords:** generalized nonlinear Schrödinger equation; Kerr nonlinearity; optical soliton; simplest equation method; dispersion of unrestricted order

**MSC:** 35A24; 35C05; 35C07; 35C08

## 1. Introduction

The investigation of the effect of high-order dispersion on the propagation of pulses in a nonlinear optical medium has been presented in several papers (see, for example, papers [1–17]). Usually, these studies were aimed at constructing optical solitons for specific high-order equations. The appearance of terms with a high order of dispersion in the generalized nonlinear Schrödinger equation is explained by taking into account the expansion of the mode propagation constant in a Taylor series around the carrier frequency [18–20]. The influence of terms with high-order derivatives is usually neglected, since the coefficients of these derivatives have smaller values compared to the coefficients for low-order derivatives. However, it is known that neglecting the influence of high-order derivatives in nonlinear mathematical models is often incorrect, since their influence appears at late times and long distances of wave propagation. In this connection, in the paper [12], a hypothesis about the form of an optical soliton for the generalized nonlinear Schrödinger equations with Kerr nonlinearity and an unrestricted order of dispersion was formulated.

In this paper, we consider the family of the generalized nonlinear Schrödinger equations in the form

$$i\, q_t + \sum_{j=1}^{n} \alpha_{2j}\, q_{2j,x} + i \sum_{j=2}^{n} \alpha_{2j-1}\, q_{2j-1,x} = \beta\, |q|^2\, q,$$

$$n \in \mathbb{N}, \qquad q_{m,x} = \frac{\partial^m q}{\partial x^m}, \qquad m \in \mathbb{N}, \tag{1}$$

where $i^2 = -1$, $q(x,t)$ is a complex function, and $t$ and $x$ are independent variables.

The following equations belong to the equations of family (1). We have the famous nonlinear Schrödinger equation at $n = 1$ [19]:

$$i\,q_t + \alpha_2\,q_{xx} = \beta\,|q|^2\,q. \tag{2}$$

Substituting $n = 2$ into (1) yields the partial differential equation of the fourth order [11]:

$$i\,q_t + \alpha_2\,q_{xx} + i\,\alpha_3\,q_{xxx} + \alpha_4\,q_{xxxx} = \beta\,|q|^2\,q. \tag{3}$$

The differential equation of the sixth order in the form [12]

$$i\,q_t + \alpha_2\,q_{xx} + i\,\alpha_3\,q_{xxx} + \alpha_4\,q_{xxxx} + i\,\alpha_5\,q_{xxxxx} + \alpha_6\,q_{xxxxxx} = \beta\,|q|^2\,q \tag{4}$$

is obtained by substituting $n = 3$ into (1), and so on.

At first glance, it seems that Equation (1) does not have any physical meaning and cannot have any physical applications. However, it should be kept in mind that the second term of the Taylor series expansion of the function $q(x, t)$ is used to take into account the effect of dispersion in the nonlinear Schrödinger equation [19]. Equation (1) is interesting in that it takes into account higher orders of dispersion when describing the propagation of a pulse in an optical medium.

The objective of this paper is to find the optical solitons of Equation (1) at all integer $n \in \mathbb{N}$ in analytical form.

The paper is organized as follows. In Section 2, we use the Painlevé test to investigate the integrability of Equation (1). Using traveling wave reduction, we obtain two arbitrary constants in the expansion of the general solution in the Laurent series. In Section 3, we prove the theorem of existence of optical solutions for the generalized nonlinear Schrödinger equation with the Kerr nonlinearity and dispersion of unrestricted order. We present the form of optical soliton for the equation with the unrestricted order of dispersion. In Section 4, we present the calculations of parameters of the equation and optical soliton of the generalized nonlinear Schrödinger equation of the twelfth order.

## 2. Application of the Painlevé Test to Equation (1)

We look for the optical solitons of Equation (1) in the form

$$q(x, t) = y(z)\,e^{i(kx + \omega\,t + \theta_0)}. \tag{5}$$

Substituting (5) into Equation (1), we obtain the imaginary part of Equation (1) in the linear form

$$\sum_{j=1}^{n} P_{2j-1}\,y_{2j-1,z} = 0 \tag{6}$$

and the real part of the nonlinear equation in the form

$$\sum_{j=1}^{n} P_{2j}\,y_{2j,z} - \beta\,y(z)^3 = 0, \tag{7}$$

where $P_{2j}$ and $P_{2j+1}$ are expressions depending on the coefficients $\alpha_{2j},\ (j = 1, \ldots, n)$ and $\alpha_{2j+1},\ (j = 0, \ldots, n-1)$.

In the next section, we demonstrate that the problem of finding optical solitons of Equation (1) is reduced to the solution of Equation (7). In this section, we apply the Painlevé test to understand the integrability of Equation (7).

It is well known that the Painlevé analysis is one of the powerful approaches for determining the integrability of nonlinear differential equations. It allows us to find the necessary conditions for the existence of a general solution of a differential equation. The application of the Painlevé test to the analysis of nonlinear differential equations consists,

as a rule, of three consecutive steps. In the first step, an equation with leading terms corresponding to Equation (7) and the number of branches of the expansion in the Laurent series are found.

Taking into account Equation (7), we obtain the equation with the leading members in the form

$$\alpha_{2n}\, y_{2n,z} - \beta\, y^3 = 0. \tag{8}$$

Equation (8) is autonomous, and the first term in the expansion of the general solution of Equation (7) in the Laurent series is determined by substituting the expression

$$y(z) = d_0\, z^p \tag{9}$$

into (8).

We obtain two branches of the expansion of the general solution of Equation (7):

$$p = -n, \qquad d_0 = \pm \sqrt{\frac{\alpha_{2n}\,(3n-1)!}{\beta\,(n-1)!}}. \tag{10}$$

In the second step, we define the Fuchs indices that can determine the arbitrary coefficients of the expansion of the general solution into a Laurent series. With this aim, we substitute the solution in the form

$$y(z) = \pm \sqrt{\frac{\alpha_{2n}\,(3n-1)!}{\beta\,(n-1)!}}\, z^{-n} + d_j\, z^{j-n} \tag{11}$$

into Equation (8) and equate the coefficients of $d_j$ to zero. As a result, we obtain the algebraic equation for the index $j$ in the form

$$E = (n-j)(n-j+1)(n-j+2)\,\ldots\,(3n-2-j)\,(3n-j-1)- \tag{12}$$

$$3\,n\,(n+1)(n+2)\,\ldots\,(3n-2)\,(3n-1) = 0.$$

From Equation (12), the two following integer Fuchs indices follow in the form

$$j_1 = -1. \qquad j_2 = 4\,n. \tag{13}$$

We cannot find the other Fuchs indices in the general case. We performed calculations for $n = 2, 3, 4, 5$, and $n = 6$ and found that remaining Fuchs indices are complex numbers. As a result, we obtain that Equation (7) does not pass the Painlevé test.

We see that there is always one arbitrary constant $z_0$ in the expansion of the solution into the Laurent series because we can shift $z \to z - z_0$. However, in the third step, we have to check the coefficient at $j = 4\,n$ in the Laurent series expansion. Unfortunately, for this step of the Painlevé test, one can only check easily for the first several values of $n$.

For example, let us consider Equation (7) at $n = 3$. It takes the form

$$a_6\, y_{zzzzzz} + a_4\, y_{zzzz} + a_2\, y_{zz} - a_0\, y - \beta\, y^3 = 0, \tag{14}$$

where the coefficients $a_6$, $a_4$, $a_2$, and $a_0$ depend on coefficients $\alpha_6$, $\alpha_4$, $\alpha_2$, $k$, and $\omega$ by formulas

$$a_6 = \alpha_6, \quad a_4 = \alpha_4 + 15\, k^2\, \alpha_6, \quad a_2 = \alpha_2 + 6\, k^2 \alpha_4 + 75\, k^4\, \alpha_6, \tag{15}$$

$$a_0 = \omega + k^2 \alpha_2 + 3\, k^4\, \alpha_4 + 35\, k^6\, \alpha_6.$$

The equation with leading members corresponding to Equation (14) can be written as

$$a_6 \, y_{zzzzzz} - \beta \, y^3 = 0. \tag{16}$$

Substituting

$$y = \frac{b_0}{z^p} \tag{17}$$

into Equation (14), we obtain two branches of the expansion of the general solution of Equation (14):

$$p = 3, \quad b_0 = \pm 24 \sqrt{\frac{35 \, a_6}{\beta}}. \tag{18}$$

Substituting the solution $y(z)$ in the form

$$y = \pm 24 \sqrt{\frac{35 \, a_6}{\beta}} \, z^{-3} + b_j \, z^{j-3} \tag{19}$$

again into Equation (16) and equating coefficients of $a_j$ to zero, we find the following Fuchs indices

$$j_1 = -1, \quad j_2 = 12, \quad j_{3,4,5,6} = \frac{11}{2} \pm \frac{\sqrt{-67 \pm 4 \, i \, \sqrt{1151}}}{2}. \tag{20}$$

We obtain that Equation (14) does not pass the Painlevé test and therefore is not integrable.

However, we need to check the arbitrary coefficient corresponding to the Fuchs index $j_2 = 12$. With this aim we use the Laurent series for the solution of Equation (14) with undetermined coefficients in the form

$$y(z) = \frac{b_0}{z^3} + \frac{b_1}{z^2} + \frac{b_2}{z} + b_3 + b_4 \, z + b_5 \, z^2 + b_6 \, z^3 + b_7 \, z^4 + b_8 \, z^5 +$$

$$\tag{21}$$

$$b_9 \, z^6 + b_{10} \, z^7 + b_{11} \, z^8 + b_{12} \, z^9 + \ldots.$$

Substituting series (21) into Equation (14), we obtain the following values of coefficients for the expansion of the solution in the Laurent series

$$b_0 = \pm 24 \sqrt{\frac{35 \, a_6}{\beta}}, \quad b_1 = 0, \quad b_2 = \frac{12 \, a_4 \sqrt{35 \, \beta a_6}}{83 \, \beta a_6}, \quad b_3 = 0, \tag{22}$$

$$b_4 = \frac{a_2 \sqrt{35 \, \beta \, a_6}}{210 \, \beta \, a_6} - \frac{1177 \, a_4{}^2 \sqrt{35 \, \beta \, a_6}}{1446690 \, \beta \, a_6{}^2}, \quad b_5 = 0, \tag{23}$$

$$b_6 = -\frac{a_0 \sqrt{35 \, \beta \, a_6}}{2520 \, \beta \, a_6} - \frac{11 \, a_2 \, a_4 \sqrt{35 \, \beta \, a_6}}{209160 \, \beta \, a_6{}^2} + \frac{967 \, a_4{}^3 \sqrt{35 \, \beta \, a_6}}{120075270 \, \beta a_6{}^3}, \quad b_7 = 0, \tag{24}$$

$$b_8 = \frac{\sqrt{35 \, \beta \, a_6} \, a_0 \, a_4}{418320 \, \beta \, a_6{}^2} - \frac{\sqrt{35 \, \beta \, a_6} \, a_2{}^2}{1058400 \, \beta \, a_6{}^2} + \frac{2857 \sqrt{35 \, \beta \, a_6} \, a_2 \, a_4{}^2}{3645658800 \, \beta a_6{}^3} -$$

$$\frac{4775989 \sqrt{35 \, \beta \, a_6} \, a_4{}^4}{50029886946400 \, \beta \, a_6{}^4}, \quad b_9 = 0, \tag{25}$$

$$b_{10} = \frac{\sqrt{35 \, \beta \, a_6} \, a_0 \, a_2}{23284800 \, \beta \, a_6{}^2} - \frac{337 \sqrt{35 \, \beta \, a_6} \, a_0 \, a_4{}^2}{13367415600 \, \beta \, a_6{}^3} + \frac{431 \sqrt{35 \, \beta \, a_6} a_2{}^2 a_4}{20292703200 \, \beta \, a_6{}^3} -$$

$$\frac{1524433 \sqrt{35 \, \beta a_6} \, a_2 \, a_4{}^3}{139796432344800 \, \beta \, a_6{}^4} + \frac{2194769053 \sqrt{35 \, \beta \, a_6} \, a_4{}^5}{1926115244846654400 \, \beta \, a_6{}^5}, \quad b_{11} = 0. \tag{26}$$

We also obtain that $b_{12}$ is an arbitrary constant. As a result, we obtain the expansion in the Laurent series with two arbitrary constants, taking into account the arbitrariness of $b_{12}$ and $z_0$ because we can change the variable $z \to z - z_0$. Therefore, Equation (7) is not integrable but this equation can have the special solution with two arbitrary constants. This fact tells us that the solution of Equation (7) can be found using the method of simplest equations [21].

### 3. Theorem of Existence for the Optical Soliton of Equation (1) with Unrestricted Dispersion

In this section, we prove that Equation (1) at any integer $n$ has the solution in the form of bright optical soliton. We formulate this fact in the form of the following theorem.

**Theorem 1.** *The function of x and t in the form*

$$q(x,t) = \frac{2^{2n} A_n \mu^n e^{i(kx+\omega t-\theta_0)}}{\left(4\,\mu\,\nu\,e^{-\sqrt{\mu}(x-C_0t-x_0)} + e^{\sqrt{\mu}(x-C_0t-x_0)}\right)^n}, \tag{27}$$

*where $A_n$, $\mu$, $\nu$, $k$, $x_0$, and $\theta_0$ are arbitrary constants and value $2n$ gives the order of equation, is a bright soliton of Equation (1) at any integer $n \in \mathbb{N}$ and certain constraints on $\alpha_j$ $(j = 1, 2, \ldots n)$, $C_0$, and $\omega$.*

**Proof.** The proof of this theorem is obtained using direct calculations.

For compatibility of the system of Equations (6) and (7) we first find the constraints on the coefficients $\alpha_{2j-1}$ $(j = n, n-1, \ldots, 2)$ and $C_0$ from the linear Equation (6). In this case, any smooth function $y(z)$ is a solution of Equation (6). Therefore, the problem of finding the solution of Equation (1) is reduced to the solution of Equation (7).

We look for the solution of Equation (7) as follows [21–25]:

$$y(z) = A_n R(z)^n, \tag{28}$$

where $R(z)$ is a solution of Equation [21]:

$$R_z^2 = \mu R^2 - \nu R^4. \tag{29}$$

Differentiating (29) with respect to $z$, we obtain

$$R_{zz} = \mu R - 2\nu R^3. \tag{30}$$

It is easy to see that all solutions of Equation (29) are also solutions of Equation (30). Taking into account the solution (28) and Equations (29) and (30), we obtain

$$y_{zz} = A_n n^2 \mu R^n - A_n \nu n(n+1) R^{n+2}, \tag{31}$$

$$y_{zzzz} = A_n \mu^2 n^4 R^n - 2 A_n \mu \nu \left(n^4 + 3n^2 + 4n^2 + 2n\right) R^{n+2} + \tag{32}$$

$$A_n \nu^2 \left(n^4 + 6n^3 + 11n^2 + 6n\right) R^{n+4}. $$

By induction we obtain the equality

$$y_{2n,z} = A_n F_n \mu^n R^n + \ldots + A_n \nu^n G_n R^{3n}, \quad y_{2n,z} = \frac{d^{2n}y}{dz^{2n}}, \tag{33}$$

where $F_n$ and $G_n$ are polynomials in $n$.

One can also note that

$$y^3 = A_n^3 R(z)^{3n}. \tag{34}$$

Taking into account (33) and (34), we can find the coefficient from Equation (7) in the form

$$a_{2n} = (-1)^n \frac{A_n^2 \beta}{G_n \nu^n}. \tag{35}$$

Then, using the value $a_{2n}$, we can find the coefficients $a_{2j}$ $(j = n-1, n-2, \ldots, 1)$ and $\omega$.

The solution of Equation (29) takes the form [21–25]

$$R(z) = \pm \frac{4\,\mu}{4\,\mu\,\nu\,e^{-\sqrt{\mu}(z-z_1)} + e^{\sqrt{\mu}(z-z_0)}}. \tag{36}$$

Substituting (36) into (28), we obtain the function (27) that is a solution of Equation (1) with constraints on the parameters of the equation. Thus, there is always solution (27) of Equation (1). □

We have to note that solution (27) of Equation (1) in the case of an unrestricted order is new. However, earlier, in papers [11,12,26], solutions were found at $n = 1$, $n = 2$, and $n = 3$. These solutions coincide with solutions obtained by formulas (27) at $n = 1$, $n = 2$, and $n = 3$. The approach of this section can also be used to study fractional differential equations considered in papers [27–29].

### 4. Optical Solitons of the Twelfth-Order Equation (1)

Let us demonstrate the application of the method for construction of solution (27) of the twelfth-order Equation (1). Assuming $n = 6$ in Equation (1), we obtain the equation in the form

$$\begin{aligned}
& i\,q_t + \alpha_2\,q_{2,x} + i\,\alpha_3\,q_{3,x} + \alpha_4\,q_{4,x} + i\,\alpha_5\,q_{5,x} + \alpha_6\,q_{6,x} + i\,\alpha_7\,q_{7,x} + \\
& \alpha_8\,q_{8,x} + i\,\alpha_9\,q_{9,x} + \alpha_{10}\,q_{10,x} + i\,\alpha_{11}\,q_{11,x} + \alpha_{12}\,q_{12,x} = \beta\,|q|^2\,q.
\end{aligned} \tag{37}$$

Substituting solution (5) into Equation (37) and equating the imaginary and real parts to zero, we obtain the system of Equations (6) and (7). The equation for the imaginary part takes the form

$$\begin{aligned}
& (12\,\alpha_{12}\,k + \alpha_{11})\,y_{11,z} + \left(10\,\alpha_{10}\,k - 220\,\alpha_{12}2\,k^3 - 55\,\alpha_{11}\,k^2 + \alpha_9\right) y_{9,z} + \\
& \left(792\,\alpha_{12}\,k^5 + 330\,\alpha_{11}\,k^4 - 120\,\alpha_{10}\,k^3 - 36\,\alpha_9\,k^2 + 8\,\alpha_8\,k + \alpha_7\right) y_{7,z} + \\
& \left(252\,\alpha_{10}\,k^5 - 792\,\alpha_{12}\,k^7 - 462\alpha_{11}k^6 + 126\alpha_9\,k^4 - 56\,\alpha_8, k^3 - 21\,\alpha_7\,k^2 + \right. \\
& \left. 6\,\alpha_6\,k + \alpha_5\right) y_{5,z} + \left(220\,\alpha_{12}\,k^9 + 165\,\alpha_{11}\,k^8 - 120\,\alpha_{10}\,k^7 - 84\,\alpha_9\,k^6 + \right. \\
& \left. 56\,\alpha_8\,k^5 + 35\,\alpha_7\,k^4 - 20\,\alpha_6\,k^3 - 10\,\alpha_5\,k^2 + 4\,\alpha_4\,k + \alpha_3\right) y_{3,z} + \\
& \left(10\,\alpha_10\,k^9 - 12\,\alpha_{12}\,k^{11} - 11\,\alpha_{11}\,k^{10} + 9\,\alpha_9\,k^8 - 8\,\alpha_8\,k^7 - 7\,\alpha_7\,k^6 + \right. \\
& \left. 6\,\alpha_6\,k^5 + 5\,\alpha_5\,k^4 - 4\,\alpha_4\,k^3 - 3\,\alpha_3\,k^2 + 2\,\alpha_2\,k - C_0\right) y_z = 0
\end{aligned} \tag{38}$$

The equation for the real part can be written as

$$\alpha_{12}y_{12,z} + \left(\alpha_{10} - 66\alpha_{12}\,k^2 - 11\alpha_{11}\,k\right)y_{10,z} + \left(495\alpha_{12}\,k^4 + 165\,\alpha_{11}\,k^3 - \right.$$
$$45\alpha_{10}\,k^2 - 9\alpha_9\,k + \alpha_8\big)\,y_{8,z} + \left(210\alpha_{10}\,k^4 - 924\,\alpha_{12}\,k^6 - 462\,\alpha_{11}\,k^5 + \right.$$
$$84\alpha_9\,k^3 - 28\alpha_8\,k^2 - 7\,\alpha_7\,k + \alpha_6\big)\,y_{6,z} + \left(\alpha_4 + 495\alpha_{12}\,k^8 + 330\,\alpha_{11}\,k^7 - \right.$$
$$210\alpha_{10}\,k^6 - 126\alpha_9\,k^5 + 70\alpha_8\,k^4 + 35\,\alpha_7\,k^3 - 15\alpha_6\,k^2 - 5\,\alpha_5\,k\big)\,y_{4,z} +$$
$$\left(45\alpha_{10}\,k^8 - 66\alpha_{12}\,k^{10} - 55\alpha_{11}\,k^9 + 36\alpha_9\,k^7 - 28\alpha_8\,k^6 - 21\,\alpha_7\,k^5 + \right.$$
$$15\alpha_6\,k^4 + 10\,\alpha_5\,k^3 - 6\,\alpha_4\,k^2 - 3\,\alpha_3\,k + \alpha_2\big)\,y_{2,z} + \left(\alpha_{12}\,k^{12} + \alpha_{11}\,k^{11} - \right.$$
$$\alpha_{10}\,k^{10} - \alpha_9\,k^9 + \alpha_8\,k^8 + \alpha_7\,k^7 - \alpha_6\,k^6 - \alpha_5\,k^5 + \alpha_4\,k^4 + \alpha_3\,k^3 - $$
$$\alpha_2\,k^2 - \omega\Big)\,y - \beta\,y^3 = 0. \tag{39}$$

From Equation (38) we obtain the constraints on the parameters of Equation (1) in the form

$$\alpha_{11} = -12\,\alpha_{12}\,k, \tag{40}$$

$$\alpha_9 = -440\,\alpha_{12}\,k^3 - 10\,\alpha_{10}\,k, \tag{41}$$

$$\alpha_7 = -12672\,\alpha_{12}\,k^5 - 240\,\alpha_{10}\,k^3 - 8\,\alpha_8\,k, \tag{42}$$

$$\alpha_5 = -215424\,\alpha_{12}\,k^7 - 4032\,\alpha_{10}\,k^5 - 112\,\alpha_8\,k^3 - 6\,\alpha_6\,k \tag{43}$$

$$\alpha_3 = -1745920\,\alpha_{12}\,k^9 - 32640\,\alpha_{10}\,k^7 - 896\,\alpha_8\,k^5 - 40\,\alpha_6\,k^3 - 4\,\alpha_4\,k \tag{44}$$

$$C_0 = 4245504\,\alpha_{12}\,k^{11} + 79360\,\alpha_{10}\,k^9 + 2176\,\alpha_8\,k^7 +$$
$$96\,\alpha_6\,k^5 + 8\,\alpha_4\,k^3 + 2\,\alpha_2\,k. \tag{45}$$

Equation (38) is satisfied for any smooth function $y(z)$ at conditions (40)–(45). We look for the solution of Equation (39) in the form

$$y(z) = A_6\,R(z)^6, \tag{46}$$

where $R(z)$ is the function (36). Substituting (36) into Equation (39) and taking into account the derivatives of $R(z)$ and conditions (40)–(45), we obtain the polynomial in $R(z)$ which has to be equal to zero. Equating the coefficients of this polynomial to zero, we find the additional constraints on the parameters of Equation (1) in the form

$$\alpha_{12} = \frac{A_6{}^2\beta}{2964061900800\,\nu^6}, \tag{47}$$

$$\alpha_{10} = -\frac{\beta\,(33\,k^2 + 398\,\mu)\,A_6{}^2}{1482030950400\,\nu^6}, \tag{48}$$

$$\alpha_8 = \frac{A_6{}^2\beta\,(165\,k^4 + 11940\,k^2\mu + 82256\,\mu^2)}{988020633600\,\nu^6}, \tag{49}$$

$$\alpha_6 = -\frac{A_6{}^2\beta\,(231\,k^6 + 41790\,k^4\mu + 1727376\,k^2\mu^2 + 9460432\,\mu^3)}{741015475200\,\nu^6}, \tag{50}$$

$$\alpha_4 = \frac{A_6{}^2 \beta\, k^8}{5988003840\, \nu^6} + \frac{199\, k^6 \mu\, A_6{}^2 \beta}{3528645120\, \nu^6} + \frac{5141\, k^4 \mu^2 A_6{}^2 \beta}{882161280\, \nu^6} +$$

$$\frac{34781\, k^2 \mu^3 A_6{}^2 \beta}{181621440\, \nu^6} + \frac{2930269\, \mu^4 A_6{}^2 \beta}{2894591700\, \nu^6}, \tag{51}$$

$$\alpha_2 = -\frac{A_6{}^2 \beta\, k^{10}}{44910028800\, \nu^6} - \frac{199\, A_6{}^2 \beta\, k^8 \mu}{16467010560\, \nu^6} - \frac{5141\, A_6{}^2 \beta\, k^6 \mu^2}{2205403200\, \nu^6} -$$

$$\frac{34781\, A_6{}^2 \beta\, k^4 \mu^3}{181621440\, \nu^6} - \frac{2930269\, A_6{}^2 \beta\, k^2 \mu^4}{482431950\, \nu^6} - \frac{3131984\, \mu^5 A_6{}^2 \beta}{80405325\, \nu^6}, \tag{52}$$

$$\omega = \frac{A_6{}^2 \beta\, k^{12}}{269460172800\, \nu^6} + \frac{199\, A_6{}^2 \beta\, k^{10} \mu}{82335052800\, \nu^6} + \frac{5141\, A_6{}^2 \beta\, k^8 \mu^2}{8821612800\, \nu^6} +$$

$$\frac{34781\, A6^2 \beta\, k^6 \mu^3}{544864320\, \nu^6} + \frac{2930269\, A_6{}^2 \beta\, k^4 \mu^4}{964863900\, \nu^6} + \frac{3131984\, A_6{}^2 \beta\, k^2 \mu^5}{80405325\, \nu^6} -$$

$$\frac{4096\, A_6{}^2 \beta\, \mu^6}{7293\, \nu^6}. \tag{53}$$

The solution of the generalized Schrödinger Equation (37) can be written as follows:

$$q(x,t) = \frac{4096\, A_6\, \mu^6\, e^{i(kx + \omega t + \theta_0)}}{\left(4\, \mu\, \nu\, e^{-\sqrt{\mu}(x - C_0 t - z_0)} + e^{\sqrt{\mu}(x - C_0 t - z_0)}\right)^6}. \tag{54}$$

One can note that $A_6$, $\mu$, $\nu$, $k$, $z_0$, and $\theta_0$ are arbitrary constants in solution (54). However, the parameters $C_0$ and $\omega$ are determined by formulas (45) and (53). The other parameters of Equation (37) are found taking into account formulas (40)–(44) and (47)–(52).

## 5. Conclusions

In this paper, we considered the generalized Schrödinger equation with Kerr nonlinearity and unrestricted order of dispersion. We applied the Painlevé test and showed that equations of this family are not integrable in the general case, and the Cauchy problem cannot be solved by the inverse scattering transform. However, we obtained that there are two arbitrary constants in the expansion of the general solution into the Laurent series and we showed there are special solutions of Equation (1). We looked for solutions of this equation using the traveling wave reduction. We proved the theorem claiming that all differential equations of this family have optical solitons in analytical form. We presented the detailed calculations for the nonlinear differential equations of the twelfth order.

**Funding:** This research was funded by Russian Science Foundation. Grant number 22-11-00141.

**Data Availability Statement:** Not applicable.

**Acknowledgments:** This research was supported by Russian Science Foundation Grant No. 22-11-00141 "Development of analytical and numerical methods for modeling waves in dispersive wave guides". The author is grateful to the anonymous reviewers for valuable remarks on the paper, contributing to its improvement. The author also declares that there are no conflicts of interest.

**Conflicts of Interest:** The authors declare no conflict of interest.

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
