# Peer review of "Optical Solitons of the Generalized Nonlinear Schrödinger Equation with Kerr Nonlinearity and Dispersion of Unrestricted Order"

_mathematics, doi:10.3390/math10183409_

Round 1

Reviewer 1 Report

Dear author,

This is the review report for the paper under title: " Optical Solitons of the Generalized Nonlinear Schrödinger Equation with Kerr Nonlinearity and Dispersion of Unrestricted Order". In my opinion, the authors discusses an interesting equation with suitable tecniques. Previous to provide an acceptance, I recommend the following comments for consideration:

1) The introduction does not locate the problem under study into a  research context. Remark the novelty.

2) The number or cited articles should be further described. Select the most relevant articles for the definition of your research and discuss them. In addition, I would recommend the author to discuss that there may exist other kinds of solutions like Travelling Waves (see the coming references that can be cited as examples), but he considered the optical soliton solutions because of negiglible dispersive effects that leads to preserve the solution shape (or other reasons to explain...):

- https://doi.org/10.3390/sym14071451

 https://doi.org/10.3390/math10101729

3) Introduce the variable "z" in the expression (5).

4) In general there are many grammar mishaps that need to be corrected.

5) In general, I recommend to provide further justifications about the proposed solutions. For example, why the author consider a solution of the form (28)?

6) I found the involved calculations correct, but I recommend to provide a more comprenhensive view of the steps followed. It seems to me that there are plenty of assessments, but not clear why the author has followed them. For example, provide further justifications about the Fluck index up to 12 in the expression (21).  Symilar comment to the Section 4 where the author studies the twelfth-order equation. 

Author Response

Reply to the first Reviewer

Remark 1. The introduction does not locate the problem under study into a research context. Remark the novelty.

Reply 1. We add a few phrases in introduction about novelty our results (page 2, lines 36-40). “As far as we know, the analytical solutions of the family of Eq. \eqref{Bas} have been obtained only for the first several values $n$. The objective of this paper is to find the optical solitons of  Eq. \eqref{Bas} at all integer $n \in \mathbb{N} $ in analytical form. Analytical solutions for optical solitons of Eq. \eqref{Bas} for arbitrary integers $n$ are obtained for the first time in this paper and are new”.

Remark 2. The number or cited articles should be further described. Select the most relevant articles for the definition of your research and discuss them. In addition, I would recommend the author to discuss that there may exist other kinds of solutions like Travelling Waves (see the coming references that can be cited as examples), but he considered the optical soliton solutions because of negiglible dispersive effects that leads to preserve the solution shape (or other reasons to explain...):

Reply 2. We add some references that are devoted to application of Eq. (1) and to discussion about other solutions of Eq. (1) (see, page 6, lines 145-155)

 “Substituting \eqref{Sol_R} into \eqref{Sol_y}, we get the function \eqref{Sol_qq}, that is a solution of Eq. \eqref{Bas} with constraints on the parameters of the equation.

As this takes place, analytical solutions \eqref{Sol_y} have the same form for the all equations \eqref{S_2}. These solutions depend only on the order of the equation $n$. As a consequence of the analytical solution \eqref{Sol_y}, we have found that the general solution of Eq. \eqref{S_2} always has two arbitrary constants in the the Laurent series expansion. Thus, we have shown that the family of Eqs.\eqref{Bas} has optical solitons in an analytical form for an arbitrary order of the differential equation. Solution \eqref{Sol_qq} of Eq. \eqref{Bas} in the case of an unrestricted order of Eq. \eqref{Bas} is new. Exact solutions at $n=1$,  $n=2$ and $n=3$ have been found first in papers \cite{A_9, H_3, H_4}. These solutions coincide with solutions obtained by formulas \eqref{Sol_qq} at $n=1$, $n=2$ and $n=3$.

Note that there is an important question about other solutions of Eq. \eqref{Bas} except traveling wave solutions. We believe that the answer to this question is negative, since Eq. \eqref{Bas} admits only two operators \eqref{Oper} and these ones lead to the traveling wave solutions. However, other forms of traveling wave solutions like studied in papers \cite{P_1, P_2} are possible. The study of this question is beyond the scope of this work and is planned in the future. The approach of this paper can be also used to study fractional differential equations considered in papers \cite{Ha_1, Ha_2, Ha_3}”.

Remark 3.  Introduce the variable "z" in the expression (5).

Reply 3. We introduced the variable “z” in the expression (5)

Remark 4.   In general there are many grammar mishaps that need to be corrected.

Reply 4. We tried to correct some grammar mishaps.

Remark 5.  In general, I recommend to provide further justifications about the proposed solutions. For example, why the author consider a solution of the form (28)?

Reply 5. We add before formula (28) the explanation in the form “It has been shown in Section \ref{Painleve}, that the general solution of Eq. \eqref{S_2} has the pole of the $n$-th order. Taking into account this fact and  the simplest equation method \cite{Ku_2, KM_1, KM_2, KM_3, KM_4}, we look for the solution of Eq. \eqref{S_2} as follows”.

Remark 6.  I recommend to provide a more comprenhensive view of the steps followed. It seems to me that there are plenty of assessments, but not clear why the author has followed them. For example, provide further justifications about the Fuchs index up to 12 in the expression (21).  Similar comment to the Section 4 where the author studies the twelfth-order equation

Reply 6. We add a discussion on the Painleve test (see, page 3, lines 90-93): “We see that there is always one arbitrary constant $z_0$ in the expansion of the general solution into the Laurent series because we can shift $z \rightarrow z-z_0$. However, to obtain the second arbitrary constant we have to check the coefficient $d_{4\,n}$ at $j=4\,n$ in the Laurent series expansion using the third step of the Painlev\'e test. Unfortunately this step of the Painlev\'e test one can check easily only for the first several values of $n$.” 

We also add in page 4, lines 95-99: “In accordance with the previously obtained integer values of Fuchs indices for the family of Eqs. \eqref{S_2} one can expect their values: $j_1=-1$ and $j_2=12$. Taking into account $n=3$ we also have to obtain $b_{12}$ as the arbitrary coefficient $”.

The author is grateful to the anonymous reviewer for benevolent and valuable comments on the text of the work, contributing to the improvement of the paper.

Reviewer 2 Report

    "Optical Solitons of the Generalized Nonlinear Schrödinger Equation with Kerr Nonlinearity and Dispersion of Unrestricted Order" is the topic of the author's work. The solutions of equations are looked for using traveling wave reductions. The Painlevé test is applied for finding arbitrary constants in the expansion of the general solution into the Laurent series. The main result of this paper is the theorem of the existence of optical solitons for equations of unrestricted order that is proved by direct calculation.

    The structure of the manuscript is clear and the analysis is acceptable.

 (1) The author has to compare his study and results with published studies, to clarify their merits. This comparison is absolutely necessary.

 (2) All references should be arranged alphabetically.

 (3) If possible, give some applications.

 (4) Correction of the title in reference 11.

    In addition to minor corrections:

  ∙  page 1, line 13: change "a number of" by "several";

  ∙  page 2, line 36: change " In Section 2" by "In Section 2, ";

  ∙  page 2, line 37: change " Using the traveling wave reduction" by "Using traveling wave reduction,";

  ∙  page 2, line 38&39: change " In Section 3" by " In Section 3,";

  ∙  page 2, line 42: change " In Section 4" by " In Section 4,";

  ∙  page 2, line 50: change "In the next section" by "In the next section, ";

  ∙  page 2, line 57: change "At the first step," by "In the first step,";

  ∙  page 4, line 100: change "In this Section" by "In this section,";

  ∙  page 8, line 137: change "In this paper" by "In this paper,";

  ∙  page 8, line 140: change "However" by "However,";

    Thanks and Best Regards.

    Yours sincerely.

Author Response

Remark 1. The author has to compare his study and results with published studies, to clarify their merits. This comparison is absolutely necessary.

Reply 1. In p. 6 (lines 130-134 ) we add the words about comparison with the pulished results “We have to note that solution \eqref{Sol_qq} of Eq. \eqref{Bas} in the case of an unrestricted order is new. However, earlier in papers \cite{A_9, H_3, H_4}, solutions were found at $n=1$, $n=2$ and $n=3$. These solutions coincide with solutions obtained by formulas \eqref{Sol_qq} at $n=1$, $n=2$ and $n=3$. The approach of this Section can be used to study of fractional differential equations considered in papers \cite{Ha_1, Ha_2, Ha_3}”

Remark 2. All references should be arranged alphabetically.

Reply 2. The references in the revised paper have been done alphabetically.

Remark 3. If possible, give some applications.

Reply 3. On the page 2 (line 34-39), we add an insertion into this variant of the paper “At first glance, it seems that Eq. \eqref{Bas} does not have any physical meaning and cannot have any physical applications. However, it should be kept in mind that the second term of the Taylor series expansion of the function $q(x,t)$ is used to take into account the effect of dispersion in the nonlinear Schr\"{o}dinger equation \cite{Kiv_2}. Eq. \eqref{Bas} is interesting in that it takes into account higher orders of dispersion when describing the propagation of a pulse in an optical medium”.

Remark 4. Correction of the title in reference 11.

Reply 4. Correction of the title in reference 11 has been done.

The minor corrections have been done as well.

The author is grateful to the anonymous reviewer for benevolent and valuable comments on the text of the work, contributing to the improvement of the paper.

Reviewer 3 Report

In this work, a general case of the Schrodinger equation in the space (t,x), is considered. The integrability conditions are investigated. Finally, a soliton solution is derived. The paper is well-written and contains novelties. However, there are some comments to be responded:

1-      In Eq. (1), the odd order of derivatives with respect to the space direction, have the imaginary unit coefficient. But, the term   in Eq. (4), has not this coefficient.

2-      In line 56 of Page 2, it is mentioned that: the Painlevé test consists of three steps. I found only two steps descriptions.

3-    The values of , in the Theorem of page 4, Eq. (27), is not defined.

4-      The last term in the general form of  in Eq. (32), is . This is inconsistent with the and , as the last terms of  and  in Eqs. (30) and (31), respectively.

5-      The reported solution (27), is a soliton solution of general Eq. (1), which is independent of the coefficients  and . In the illustrative example, one can found that the values of are very restrictive. Is it possible to derive some more generalized solutions with the current method?

6-      I recommend to discus more about the exact solutions of differential equations by the techniques mentioned in the following works:

-https://doi.org/10.1016/j.ijleo.2017.03.094

- https://doi.org/10.1016/j.rinp.2022.105512

- https://doi.org/10.1007/s40314-022-01977-1

Author Response

Remark 1. In Eq. (1), the odd order of derivatives with respect to the space direction, have the imaginary unit coefficient. But, the term   in Eq. (4), has not this coefficient.

Reply 1. Correction in Eq. (1) has been done. It was misprint. Thanks a lot.

Remark 2. In line 56 of Page 2, it is mentioned that: the Painlevé test consists of three steps. I found only two steps descriptions.

Reply 2. The third step has not been done because the second step showed that the study can be stopped. The equation does not pass the Painlevet test. A corresponding explanation has been added to this text on page 3 (lines 81-84 ) “We see that there is always one arbitrary constant $z_0$ in the expansion of the solution into the Laurent series because we can shift $z \rightarrow z-z_0$. However, in the third step we have to check the coefficient at $j=4\,n$ in the Laurent series expansion. Unfortunately this step of the Painlev\'e test one can check easily only for the first several values of $n$.”

Remark 3. The values of , in the Theorem of page 4, Eq. (27), is not defined.

Reply 3. We have defined some additional parameters in the theorem of page 4.

Remark 4. The last term in the general form of  in Eq. (32), is . This is inconsistent with the and , as the last terms of  and  in Eqs. (30) and (31), respectively.

Reply 4. I checked and corrected formulas (30) and (31).

Remark 5. The reported solution (27), is a soliton solution of general Eq. (1), which is independent of the coefficients  and . In the illustrative example, one can found that the values of are very restrictive. Is it possible to derive some more generalized solutions with the current method?

Reply 5. The question about some more generalized solution is interesting and important but my opinion is we need to consider this question in future. In fact the author believes that there are some other solutions for other constraints of parameters. However I am sure there is not the general solution because the equation does not pass the Painlev\’e test because we have shown in this paper.   

Remark 6.  I recommend to discus more about the exact solutions of differential equations by the techniques mentioned in the following works.

Reply 6. It has been done. See page 6 (lines 30-134) in this version of paper.  

The author is grateful to the anonymous reviewer for benevolent and valuable comments on the text of the work, contributing to the improvement of the paper.

Round 2

Reviewer 1 Report

Accept

Reviewer 2 Report

The author has made all the modifications and in light of this, we can accept the publication of the manuscript.

Reviewer 3 Report

I recommend for publication in the present form.